# Antihypertensive and Probiotic Effects of Hidakakombu (*Saccharina angustata*) Fermented by *Lacticaseibacillus casei* 001

**DOI:** 10.3390/foods10092048

**Published:** 2021-08-31

**Authors:** Tetsuya Sekine, Hiroshi Nagai, Naoko Hamada-Sato

**Affiliations:** 1Course of Safety Management in Food Supply Chain, Tokyo University of Marine Science and Technology, Konan-4, Minato-ku, Tokyo 108-8477, Japan; s.tetsuya1125@gmail.com; 2Department of Ocean Sciences, Tokyo University of Marine Science and Technology, Konan-4, Minato-ku, Tokyo 108-8477, Japan; nagai@kaiyodai.ac.jp; 3Department of Food Science and Technology, Tokyo University of Marine Science and Technology, Konan-4, Minato-ku, Tokyo 108-8477, Japan

**Keywords:** Hidakakombu, lactic acid bacteria, angiotensin-converting enzyme inhibitory activity, probiotics, D-amino acids

## Abstract

Hidakakombu (*Saccharina angustata*), commonly known as kelp, is an edible macroalgae mainly grown in the Hidaka region of Hokkaido. Hidakakombu is graded based on its shape and color. Low-grade Hidakakombu has low value and is distributed at a low price. It is desired to establish a method to add value to low-grade Hidakakombu. In this study, low-grade Hidakakombu was fermented by *Lacticaseibacillus casei* 001 to add value. Fermentation of Hidakaombu enhanced the inhibition of blood pressure elevation due to ACE inhibition. *L. casei* 001 in fermented Hidakakombu remained viable in simulated gastric and intestinal juices. The ACE inhibitory component in *L. casei* 001-fermented Hidakakombu was isolated from the fraction below 3 kDa using high-performance liquid chromatography. The purified amino acid was identified as D-Trp using nuclear magnetic resonance, mass spectroscopy, and optical rotation measurements. This is the first report on the ACE inhibitory activity of D-Trp in *L. casei* 001-fermented Hidakakombu. Hidakakombu fermented by *L. casei* 001 was shown to be a source of probiotics and functional components against hypertension. Therefore, fermentation by *L. casei* 001 was found to be an effective means of adding high value to low-grade Hidakombu.

## 1. Introduction

Seaweeds are an important living resource with unique living conditions; adapted to high salinity, high pressure, low temperature, low nutrition, and other adverse environmental conditions, with different protein composition and amino acid sequence from those of terrestrial organisms [1]. Macroalgae live in highly competitive and harsh environments, and such conditions require the production of highly specific and potent bioactive compounds that may lead to the development of functional food or nutraceuticals [2]. Therefore, seaweeds have attracted attention for their functional components. The antioxidant effect of Nori [3] and the suppression of elevated blood pressure in Wakame [4] have been reported. Hidakakombu, also known as kelp, is an edible macroalgae that is mainly grown in the Hidaka region of Hokkaido. It has been used as a traditional Japanese food for a long time and is widely used in soup stock, Tsukudani (food boiled down in soy sauce), and Kombu rolls. Hidakakombu is graded based on its shape and color. Low-grade Hidakakombu has low value and is distributed at a low price. More than 80 percent of the total production of Hidakakombu is low-grade Hidakakombu. It is desired to establish a method to add value to low-grade. Anti-tumor activity [5] and anti-viral activity [6] have been reported as health functionalities of Hidakakombu. Fermentation is an ancient and widely used method for the preservation of perishable foods, hereby producing new foods [7]. The process makes use of the microbial conversion of sugars to acids such as lactic acid, acetic acid and propionic acid or to ethanol [7]. For fermentation of foods, lactic acid bacteria are the most commonly used and characterized microorganisms [8]. In fermentation using seaweeds, the antihypertensive effect of fermented laver [9] and antioxidant activity of *Sargassum* sp. [10] have been reported. In addition, Lactic acid fermentation promotes health benefits by increasing the content of probiotic bacteria [11]. However, there is no report that mentions the probiotic effect of seaweed fermentation on simulated digestive juice test. This study was focused on the fermentation of kelp with lactic acid bacteria. The antihypertensive effect enhanced by fermentation and the probiotic effect added by fermentation were examined for adding value to low-grade Hidakakombu.

Hypertension is one of the major risk factors for cardiovascular disease, with an increasing prevalence in developed countries [12]. It is estimated that approximately 25% of the world’s adult population (1.56 billion people) are affected, and the incidence is projected to increase by 29% by 2025 [12]. Hypertension is characterized by a systolic blood pressure greater than 140 mmHg and diastolic blood pressure greater than 90 mmHg [13]. In the renin-angiotensin and kallikrein-kinin systems, the angiotensin-converting enzyme (ACE) is an important factor in regulating blood pressure. Angiotensin II, which is produced by an increase in the renin-angiotensin system, constricts vascular smooth muscle and the export veins of the kidneys, which increases blood pressure [14]. Angiotensin II is produced by the action of ACE on angiotensin I; therefore, inhibiting the action of ACE prevents the production of angiotensin II and suppresses an increase in blood pressure [14]. In addition, ACE inactivates bradykinin in the kallikrein-kinin system; since bradykinin lowers blood pressure by dilating blood vessels through arterial relaxation, and inactivation of bradykinin prevents a decrease in blood pressure [14]. Therefore, inhibition of ACE contributes to the suppression of blood pressure elevation. Captopril and enalapril, which are drugs that lower blood pressure by inhibiting ACE, are potent for that purpose, but they are associated with side effects, such as increased blood pressure, cough, kidney failure, and taste disorder [15,16]. Therefore, non-pharmaceutical ingredients with ACE inhibitory activity derived from food have attracted attention; ACE inhibitory ingredients have been found in walnuts [17], wheat bran [3], mushrooms [18], and tamogitake [19].

Probiotics are defined as live microorganisms that, when administered in sufficient quantities, confer health benefits to the host [20]. Requirements for probiotics include (i) safety for humans, (ii) the presence of a sufficient number of bacteria in the fermented product to produce a probiotic effect, (iii) survival under acidic conditions in the stomach or in the presence of bile in the intestine after ingestion, and (iv) growth in the intestinal tract [21]. Therefore, it is important for probiotics to grow after passing through the stomach alive and reaching the intestinal tract, and to be resistant to gastric and intestinal juices, including stomach acid and bile, which have strong bactericidal effects. The production of organic acids, such as lactic acid and acetic acid by probiotics lowers the pH of the intestine, thereby inhibiting the growth of pathogens [22]. These organic acids also indirectly eliminate pathogens by increasing peristalsis, thereby accelerating the rate of transit through the intestine [22]. Besides these beneficial effects, prevention of diarrheal diseases [23] and enhancement of immunity [24] have also been reported. The growing interest in probiotics and increasing consumer demand has stimulated research related to this field since 2000, with more than 1000 articles and reviews on probiotics published and more than 2000 probiotic products launched [25]. In Japan, food with probiotics from lactic acid bacteria are distributed in various forms, such as yogurt and kimchi, with the keywords “Reach the intestines alive”.

In this study, the antihypertensive and probiotic effects of kelp fermented by *Lacticaseibacillus casei* 001 were examined, and an ACE inhibitory activity assay and simulated gastric and intestinal juice tolerance tests were conducted. In addition, the ACE inhibitory components in kelp fermented by *L. casei* 001 were isolated and identified. The purpose of this study is to analyze the potential health functions of the *L. casei* 001-fermented Hidakakombu and to find a means of adding value to the low-grade Hidakakombu.

## 2. Materials and Methods

### 2.1. Materials and Reagents

Hidakakombu was obtained from Yokoi Kombu (Tokyo, Japan) by drying the washed-up Hidakakombu, grinding it in a stone mortar, and then grinding it into powder (Hidakakombu powder). The kelp used in this study was the lowest value “Grade 6”. *L. casei* 001 was used from our laboratory stocks. KH_2_PO_4_, Na_2_HPO_4_, HCl, NaOH, trypsin, pancreatin, bile powder, and NaCl were purchased from Fujifilm Wako Pure Chemicals Co. (Osaka, Japan). Cellulase was purchased from Yakult Pharmaceutical Industries, Ltd. (Tokyo, Japan). Pepsin was purchased from Nacalai Tesque (Kyoto, Japan). D_2_O was purchased from Kanto Chemical Company (Tokyo, Japan), and the ACE-kit WST was purchased from Dojin Chemical Research Institute (Kumamoto, Japan). Plate Count Agar with BCP “*Nissui*” used for viable cell counting was purchased from Nissui Pharmaceutical Co. (Tokyo, Japan). Acetonitrile (ACN) was purchased from Sigma-Aldrich (St. Louis, MO, USA).

### 2.2. Fermentation of Hidakakombu by L. casei 001

Hidakakombu powder (5 g) was added to 100 mL 33 mM phosphate buffer solution (33 mM KH_2_PO_4_, Na_2_HPO_4_, pH 5.0) and adjusted to pH 5.0 using 1 N HCl. Then, 50 mg cellulase was added and incubated at 45 °C and 120 rpm for 24 h. The incubated Hidakakombu was adjusted to pH 6.8 with 1 mM NaOH and sterilized by pressure and heat (121 °C, 15 min). Hidakakombu was inoculated with 1 mL of *L. casei* 001 (10^9^ CFU/mL) pre-cultured in ionic liquids (ILS) medium at 37 °C for 1 day, followed by incubation at 37 °C. Cultures were sampled daily (0–5 days) for viable cell counts and pH measurements. Viable cell counts were measured by diluting the samples and using Plate Count Agar with BCP “*Nissui*”. The culture was lyophilized to produce *L. casei* 001-fermented Hidakakombu powder.

### 2.3. ACE Inhibitory Activity Assay

*L. casei* 001-fermented Hidakakombu powder was dissolved in H_2_O at 10 mg/mL and extracted with water (50 °C, 125 spm, 1 h). The extracted sample was centrifuged (5 °C, 8000× *g*, 10 min). The supernatant was lyophilized to obtain the ACE inhibitory activity assay sample (aqueous extract of *L. casei* 001-fermented Hidakakombu). ACE inhibitory activity was measured according to the protocol of the ACE-kit WST (Dojin Chemical Research Institute, Kumamoto, Japan), and the final concentration of the sample was set at 0.83 mg/mL. A microplate reader was used to measure absorbance.

### 2.4. Simulated Gastric Juice Tolerance Test

Ten percent of the total amount of *L. casei* 001-fermented Hidakakombu powder was added to ILS medium adjusted to pH 2.0 and pH 3.5 using 1 N HCl, and pepsin was added (final concentration: 0.04%). This was readjusted to a predetermined pH using 1 N HCl and incubated at 37 °C. Samples were tested over time (0, 1, 2, and 4 h) and used for viable cell count measurements. Viable cell counts were measured by serial dilution of the samples and plating on Plate Count Agar with BCP “*Nissui*”.

### 2.5. Simulated Intestinal Juice Tolerance Test

*L. casei* 001-fermented Hidakakombu powder exposed to simulated gastric juice (pH 3.5) at 37 °C for 2 h was added to ILS medium adjusted to 0.2% bile end concentration in 1% of the total volume. Next, Trypsin and pancreatin were added (final concentration: 0.01%). This was adjusted to pH 7.0 using 1 M NaOH and maintained at 37 °C for incubation. The culture was grown anaerobically using an Anaerobic Kenki (Mitsubishi Gas Chemical Company, Inc., Tokyo, Japan). The samples were sampled over time (0, 9, 12, 15, and 18 h) and used for viable cell count measurements. Viable cell counts were measured by serial dilution of samples and plating on Plate Count Agar with BCP “*Nissui*”.

### 2.6. Purification and Structure Determination of ACE Inhibitor Components

*L. casei* 001-fermented Hidakakombu aqueous extract was fractionated into four fractions (>30 kDa, 30–10 kDa, 10–3 kDa, and <3 kDa) using Vivaspin 20 columns (Sartorius Stedim Lab Ltd., Goettingen, Germany). The four fractions were lyophilized and subjected to ACE inhibitory activity assays. The highest ACE inhibitory activity fraction (<3 kDa) was subjected to high-performance liquid chromatography (HPLC) analysis, wherein samples dissolved in H_2_O were filtered through a 0.22 μm polyvinylidene difluoride membrane and separated on an ODS-120T column (7.8 × 300 mm) (TOSOH, Inc., Tokyo, Japan). The mobile phase used in the gradient elution consisted of distilled water (eluent A) and ACN (eluent B). The elution conditions were a linear gradient for 0–50 min of 5% eluent B, 50–51 min of 5–70% eluent B, and 51–70 min of 70% eluent B, at a flow rate of 1.0 mL/min; the peaks were detected at 220 nm. The collected peak was concentrated by a rotary evaporator and centrifugal concentrator for the ACE inhibitory activity assay. The fraction with the highest ACE inhibitory activity (Fr. B) was further purified using the same elution conditions, and the peak collected after purification was concentrated by a rotary evaporator for the following analysis. Mass spectroscopy (MS) and nuclear magnetic resonance (NMR) were used to determine the structure of the purified fraction. For MS, 10 µg of the purified fraction was dissolved in H_2_O and measured using a Bruker micrOTFO-Q II (Billerica, MA, USA). For NMR, 1 mg of the purified fraction was dissolved in 0.55 mL of D_2_O and added to an NMR tube. ^1^ H NMR,^13^ C NMR,^1^ H− ^1^ H COSY, HSQC, and HMBC spectra were measured using a Bruker ASCEND^™^ 600 MHz (Billerica, MA, USA). The optical structure of the structurally determined fractions was determined by optical rotation measurements. The purified fractions were dissolved at 2.5 mg/mL, and the angle of rotation was measured using a Jasco optical rotation meter P-2100 (Tokyo, Japan). The optical structure of the purified product was analyzed using the calculated specific optical rotation.

## 3. Results

### 3.1. Fermentation of Hidakakombu by L. casei 001

The daily changes in viable cell counts, pH, and ACE inhibitory activity of *L. casei* 001-fermented Hidakakombu are shown in Figure 1. ACE inhibitory activity increased significantly after the first day of incubation, and the pH gradually decreased. Since the viable cell counts peaked at 2 d of incubation, the 2-d culture was used as *L. casei* 001-fermented Hidakakombu for subsequent experiments (simulated gastric and intestinal juice tolerance tests and purification of ACE inhibitory components). The ACE inhibitory activity increased from 32.0% to 73.2% at a final concentration of 0.83 mg/mL by incubation at 37 °C for 2 d.

### 3.2. Simulated Gastric and Intestinal Juice Tolerance Tests

The viable cell counts of simulated gastric and intestinal juice tolerance tests of *L. casei* 001-fermented Hidakakombu are shown in Figure 2. In simulated gastric juice (pH 2.0), all *L. casei* 001 were killed by the exposure of *L. casei* 001-fermented Hidakakombu to pH 2.0. In the simulated gastric juice (pH 3.5), the viable cell counts of *L. casei* 001 were reduced, but more than 10^4^ CFU/mL remained. In addition, the viable cell counts reached 10^8^ CFU/mL after 12 h exposure to simulated intestinal juice. Under simulated intestinal juice conditions with a bile end concentration of 0.2%, the viable cell counts of *L. casei* 001-fermented Hidakakombu increased.

### 3.3. Purification and Structure Determination of ACE Inhibitor Components

The aqueous extract of *L. casei* 001-fermented Hidakakombu was fractionated into four fractions (>30 kDa, 30–10 kDa, 10–3 kDa, and <3 kDa). The IC_50_ values of all fractions were calculated to evaluate ACE inhibitory activity. The results of molecular weight fractionation are shown in Table 1. Fractions less than 3 kDa showed the strongest ACE inhibitory activity. Therefore, fractions of less than 3 kDa were used for HPLC and separated using an ODS-120T column (7.8 × 300 mm) (TOSOH, Inc., Tokyo, Japan) into three fractions (Fr. A to Fr. C) with absorption at 220 nm.

All fractions were evaluated for ACE inhibitory activity by calculating IC_50_ values; HPLC results are shown in Figure 3 and Table 2. Among these fractions, Fr. B showed the strongest ACE inhibitory activity (IC_50_ = 0.073 mg/mL). The MS spectra are shown in Figure 4. Fr. B was isolated as a white powder. The molecular formula of Fr. B was determined as C_11_H_12_N_2_O_2_ based on the ion peaks observed in the negative mode mass spectra, *m*/*z* 203.0829 ([M−H]^−^, Δ + 0.3 mmu), and positive mode spectra, *m*/*z* 205.0963 ([M + H]^+^, Δ − 0.9 mmu). The NMR spectra and data assignments are shown in Figure 5 and Table 3. Analysis of the ^1^H NMR and HSQC spectra revealed the existence of five aromatic methines (7.2–7.8 ppm), one methylene (3.1–3.5 ppm), and one methine (3.96 ppm) moieties in the molecule. ^1^ H–^1^ H COSY and HMBC analyses revealed that Fr. B was an α-amino acid tryptophan (Trp). The optical rotation [α]_D_^20^ of Fr. B was +40° (*c* = 2.5 mg/mL, H_2_O). Thus, Fr. B was determined as D-Trp.

## 4. Discussion

### 4.1. Fermentation of Hidakakombu by L. casei 001

The ACE inhibition rate increased from 32.0% to 73.2% at a final concentration of 0.83 mg/mL after incubation of Hidakakombu with *L. casei* 001 for 2 d at 37 °C. Therefore, fermentation of Hidakakombu by *L. casei* 001 potentially enhanced the inhibition of elevated blood pressure. Most studies on the functionality of seaweeds report that the components are broken down by hydrolytic enzymes to enhance their function; peptides from *Undaria pinnatifida* [26], *Palmaria palmata* [27], and *Gracilariopsis lemaneiformis* [28] are hydrolyzed peptides that inhibit ACE. However, the present study is a new report on the enhancement of ACE inhibitory activity by Hidakakombu due to microbial action. The fermentation technology in this study has the potential to be applied not only to kelp, but to many other algae, and has the advantage of providing other functional factors, such as probiotics, which will be described later, compared to the production of functional components by hydrolysis.

### 4.2. Simulated Gastric and Intestinal Juice Tolerance Tests

The pH of the stomach is maintained at approximately 2.0 and increases to 4.0 when food is ingested [29] due to the neutralization of gastric acid by the ingested food; as the food is digested, the pH decreases again. Therefore, the pH of simulated gastric juice was set at pH 2.0 and pH 3.5, assuming fasting and food intake in the stomach. Although it is difficult to set the bile end concentration as a constant value because the amount of bile secreted in the intestine depends on the type of food ingested and varies depending on the section of the gastrointestinal tract, the highest value of bile concentration in the intestine is 2% [30]. However, when converted to bile end concentration, the highest value of bile is equivalent to 0.2% [30]; thus, the bile end concentration in the simulated intestinal juice was set to 0.2%. The probiotic effect of *Lactobacillus reuteri* PUHM1004 [31] and *Lactobacillus plantarum* LRCC5193 [32] has been verified by simulated gastric and intestinal juice tolerance tests. In these studies, the probiotic effect was suggested on the basis that lactic acid bacteria survived in simulated gastric juice with low pH and pepsin and proliferated in simulated intestinal juice with bile and digestive enzymes. In this study, *L. casei* 001 maintained a viable cell counts of 10^4^ CFU/mL in simulated gastric juice (pH 3.5 pepcine 0.04%) and reached a viable cell counts of 10^8^ CFU/mL in simulated intestinal juice (bile 0.2%, trypsin 0.01%, and pancreatin 0.01%). Therefore, it was suggested that *L. casei* 001 had a probiotic effect, passing through the stomach and growing in the intestine. Seaweeds contain hydrophilic colloids such as agar, alginate, and carrageenan. These act as cryoprotectants that minimize the damage that occurs during the freezing process [33]. In this study, *L. casei* 001 was frozen and lyophilized with Hidakakombu, which may have maintained the viable cell counts in *L. casei* 001-fermented Hidakakombu powder. Since lactic acid bacteria are damaged under acidic conditions in the stomach, it is desirable to have a high initial viable cell counts. From the viewpoint of the cryoprotective effect of the hydrophilic colloid of Hidakakombu, the combination of Hidakakombu and *L. casei* 001 was suggested to be useful as a probiotic food.

### 4.3. Purification and Structure Determination of ACE Inhibitory Components

Advances in analytical technology have facilitated the separation and analysis of trace amounts of D-amino acids from L-amino acids in samples, and it has become increasingly clear that a variety of D-amino acids exist in free and bound states in many biological cells and food materials, and their physiological functions are attracting attention [34]. Recently developed analytical techniques have reported the presence of D-amino acids in various food, such as oysters, milk [35], vegetables, and fruits [36]. In particular, fermented food has been reported to contain several D-amino acids, including D-Ala, D-Asp, and D-Glu in fish sauce [37], D-Ala, D-Glu, and D-Lys in wine [38], and D-Ala, D-Asp, and D-Glu in beer [39]. It is also likely that these D-amino acids are produced by microorganisms during fermentation [38]. In this study, D-Trp (IC_50_ = 0.073 mg/mL) was isolated and purified from *L. casei* 001-fermented Hidakakombu by molecular weight fractionation and HPLC. This study is the first to report the isolation of D-Trp as an ACE inhibitory component. Trp is an essential plant-derived amino acid required for the biosynthesis of proteins in vivo [40]. After consumption, it is metabolically converted to bioactive metabolites, such as serotonin, melatonin, kynurenine, and the vitamin niacin (nicotinamide), which affects vasoconstriction, intestinal motility, primary hemostasis, liver repair, and regulation of the T-cell immune system [40,41]. Many functional properties of Trp have been reported, such as inhibition of elevated blood glucose levels [41] and anti-stress effects [42]; inhibition of elevated blood pressure has been reported by a single dose study of L-Trp in Stroke-Prone Spontaneously Hypertensive Rats [41]. L-Ile-L-Trp, L-Trp-L-Leu, L-Glu-L-Trp have been reported as an ACE inhibitory peptide containing L-Trp [43]. D-Trp, an isomer of L-Trp, has a strong sweet taste characteristic [44]. Therefore, it was hypothesized that D-Trp in fermented Hidakakombu might be involved, not only in its inhibitory effect on blood pressure elevation but also in the functional and taste properties mentioned above. Many studies have been conducted on the production of D-amino acids by lactic acid fermentation; lactic acid fermentation increases the content of D-amino acids in beer and tomato vinegar [38,45]. It is believed that isomerases, such as racemase, an extracellular enzyme metabolized from lactic acid bacteria, are involved in the production of these D-amino acids [46]. A racemase that produces D-Leu, D-allo-Ile, and D-Val from *Lactobacillus otakiensis* JCM 15040 has been purified and identified from bacterial cells [46]. Under similar HPLC conditions, unfermented Hidakakombu and fermented Hidakakombu were analyzed. The peak area of D-Trp was expanded by fermentation. This suggests that D-Trp may have been removed from Hidakakombu proteins and peptides by the action of isomerases, such as racemase, from the bacteria during fermentation. Analysis of the functional components produced by fermentation may contribute to the elucidation of the relationship between D-amino acids and fermented foods.

On the other hand, D-Trp accounts for a very small percentage of the total aqueous extract of *L. casei* 001-fermented Hidakakombu. It is likely that other compounds also contribute to the ACE inhibitory activity of *L. casei* 001-fermented Hidakakombu. In this study, D-Trp was isolated and identified in *L. casei* 001-fermented Hidakakombu, one of the ACE inhibitory components. This is the first report on the ACE inhibitory activity of D-Trp in *L. casei* 001-fermented Hidakakombu.

## 5. Conclusions

In this study, we confirmed the enhancement of ACE inhibitory activity by the action of *L. casei* 001 on low-grade Hidakakombu and the production of ACE inhibitory component (D-Trp) by fermentation. In addition, *L. casei* 001 showed in vitro digestion tolerance, indicating the probiotic effect of *L. casei* 001-fermented Hidakakombu. Lactic acid bacteria are widely accepted as a supplement to food products, being used in various food fermentation processes to produce products such as yogurt, soy sauce, and pickles. Hidakakombu fermented by *L. casei* 001 was shown to be a source of probiotics and functional components against hypertension. Therefore, fermentation by *L. casei* 001 was found to be an effective means of adding high value to low-grade Hidakakombu.

## Figures and Tables

**Figure 1 foods-10-02048-f001:**
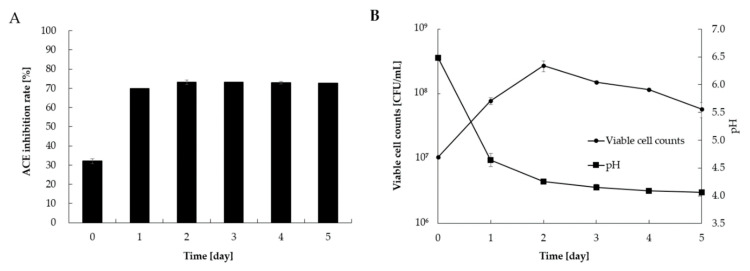
Angiotensin-converting enzyme (ACE) inhibition rate (**A**), and viable cell counts of *L. casei* 001 and pH (**B**) of *L. casei* 001-fermented Hidakakombu. (Final concentration: 0.83 [mg/mL]) (mean ± standard deviation (SD), *n* = 4).

**Figure 2 foods-10-02048-f002:**
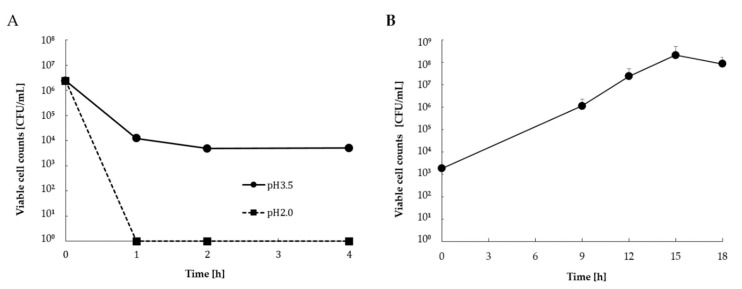
Viable cell counts of *L. casei* 001- fermented Hidakakombu in simulated gastric (**A**) and intestinal (**B**) juices (mean ± standard deviation (SD); *n* = 4).

**Figure 3 foods-10-02048-f003:**
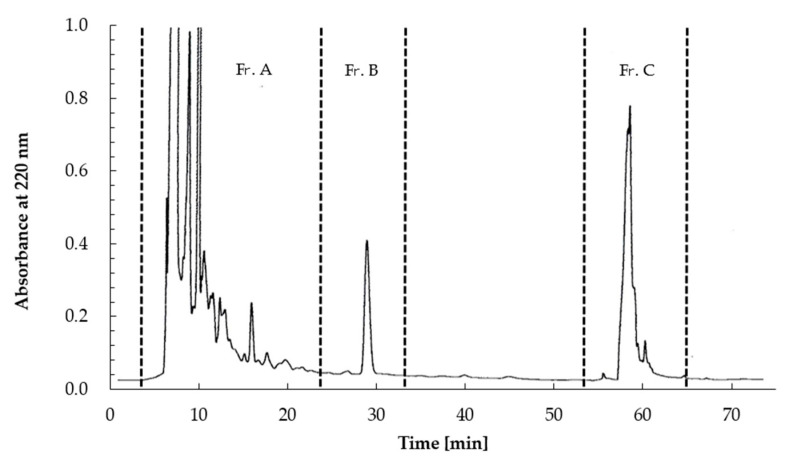
TSKgel ODS-120T filtration chromatogram of the <3 kDa fraction of *L. casei* 001-fermented Hidakakombu. Chromatogram shows three fractions (Fr. A–Fr. C).

**Figure 4 foods-10-02048-f004:**
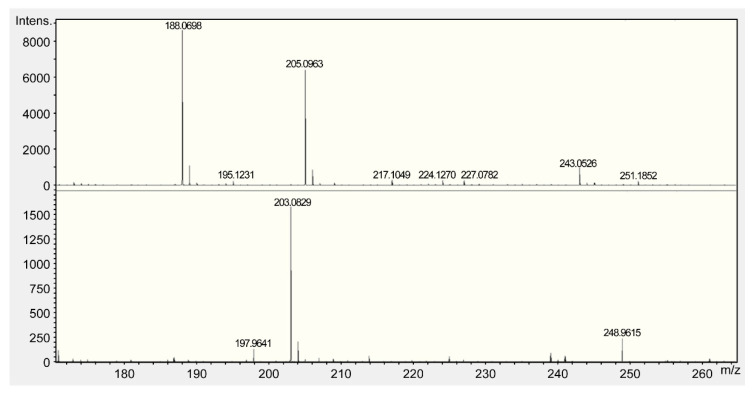
Mass spectroscopy spectrum of D-tryptophan (Trp).

**Figure 5 foods-10-02048-f005:**
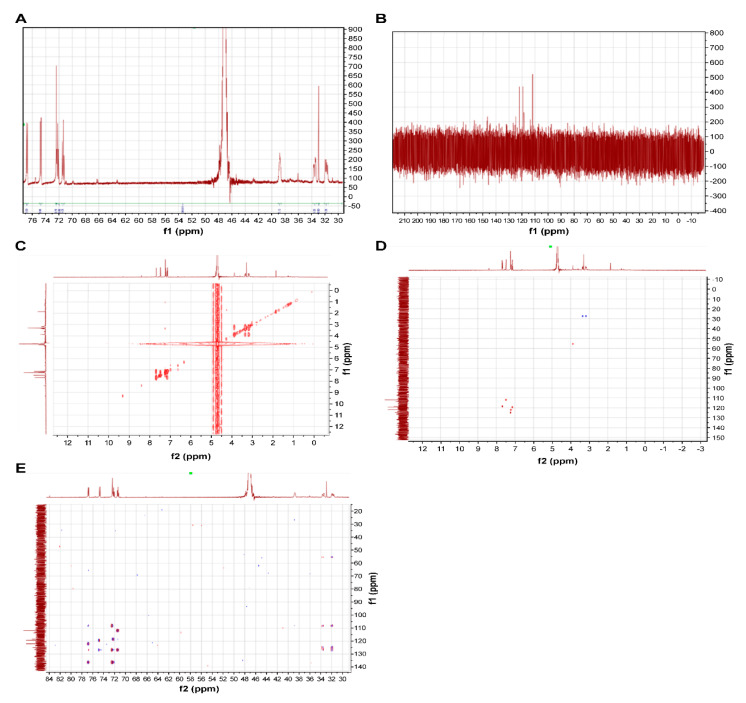
Nuclear magnetic resonance (NMR) spectra of D-tryptophan (Trp). ((**A**) ^1^ H NMR, (**B**) ^13^ C NMR, (**C**) ^1^ H–^1^ H COSY, (**D**) HSQC, (**E**) HMBC).

**Table 1 foods-10-02048-t001:** IC_50_ (Angiotensin-converting enzyme inhibitory activity) of. *L. casei* 001-fermented Hidakakombu by molecular weight.

Molecular Weight Fraction [kDa]	IC_50_ [mg/mL]	Percentage Content [%]
>30	2.24	22.48
10–30	N.D	0.04
3–10	0.67	3.26
<3	0.16	74.21

N.D: Not determined.

**Table 2 foods-10-02048-t002:** IC_50_ (Angiotensin-converting enzyme inhibitory activity) and percentage of each fraction isolated by TSKgel ODS-120T.

Fraction	IC_50_ [mg/mL]	Percentage Content [%]
Fr. A	0.35	99.0
Fr. B	0.073	0.37
Fr. C	0.18	0.33

**Table 3 foods-10-02048-t003:** NMR data assignments of D-tryptophan (Trp).

No.	δC	δH (Multiplicity)
COOH	176.40 (quaternary)	
α	55.51, CH	3.96 (dd, *J* = 4.1, 6.7)
β	27.48, CH2	3.26 (dd, *J* = 7.9, 15.3)/3.44 (dd, *J* = 4.3, 15.0)
2	124.85, CH	7.32 (s)
3	108.11 (quaternary)	
3a	136.29 (quaternary)	
4	118.53, CH	7.77 (d, *J* = 8.0)
5	122.04, CH	7.30 (dd, *J* = 7.2, 7.6)
6	119.37, CH	7.22 (dd, *J* = 7.0, 7.8)
7	111.89, CH	7.56 (d, *J* = 8.2)
7a	126.80 (quaternary)	

^13^C NMR (150 MHz), ^1^H NMR (600 MHz) in deuterium oxide.

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
