# Peer review of "Antihypertensive and Probiotic Effects of Hidakakombu (Saccharina angustata) Fermented by Lacticaseibacillus casei 001"

_foods, 2021, doi:10.3390/foods10092048_

Round 1
Reviewer 1 Report
Comments and suggestions for authors are included in the attached file.
The article has potential. An interesting topic is addressed, but there is still a significant deficiency since the article does not identify what contributes to the ACE inhibitory effect (what peptide/compound has this effect..only one amino acid is known, the 99% fraction with ACE inhibitory effect is not identified).
The antihypertensive effect mentioned in the title is only potential, because it has not been verified.
-Overall the article is written in clear language, but some sentences would benefit from linguistic correction.
-Abstract needs to be improved
Line 16-17 Please revise this sentence. The second part of the sentence communicates the same as the first.
Line 17 - The word "necessary" is not the best choice here. It is not a necessity. Rather, the point is that a method of adding value to low-grade Hidakakombu is being sought. And fermentation may constitute it..
Line 18 From 2020, the correct name of Lactobacillus casei is Lacticaseibacillus casei.
Line 18-19 Sentence should be revised
Line 23-25 This sentence should be reframed.
Introduction:
Line 38-40 The sentence needs to be revised. I suggest splitting it into 2 parts.
Line 44-45 As above, please revise this sentence.
Line 46 Redundant repetition of a word Hidakakombu.
Line 46-47 Sentences are not properly interconnected.
The introduction should include at least one sentence linking Hidakakombu to probiotics, e.g. regarding kelp fermentation with probiotic bacteria.
Materials and methods:
Line 108, 109 - 1/30 M is not an appropriate way to describe the molarity of a solution and its individual components.
Line 112 The amount of L.casei cells should be given in CFU, as this 1% does not explain how many bacterial cells were present in this volume.
Line 116 What cryoprotectant was used during lyophilisation?
Line 146-142 Simulated gastric and intestinal juice tolerance tests could simply be numbered as sections 2.4 and 2.5
Line 119-120 This sentence is not entirely clear - as I understand it, aqueous extraction was just after dissolving in water, not after centrifugation?
Line 124 Why was the concentration of 0.83 mg/mL chosen?
Line 129-130 Perhaps it would be better to simply state the final pepsin concentration (1%) here? The initial concentration is not that important..
Line 137, 138 - The authors mean 1% of the total volume of..? This is not entirely clear.
Results
Line 177 It would be beneficial to clarify what concentration the authors are referring to.
Line 180 Viable cell counts of L. casei 001 and pH is indicated as A and is in Figure B. Conversely for the graph for ACE inhibition rate.
Line 227 Please consider putting NMR spectra in supplementary material (there they could have a larger size and would be readable). Here they are not necessary.
Line 231 Figure 6 with Trp structure is unnecessary.
- Why did the authors analyze only Fr. B from the <3 kDa fraction of L. casei 001-fermented Hidakakombu? I understand that it had the highest ACE- inhibitory activity, but it accounted for 0.37% of the < 3kDa fraction. So overall anyway Fr A accounting for 99% must have contributed to the inhibitory effect of <3 kDa fraction. Only one amino acid (Trp) from the whole mixture that causes ACE inhibition is recognized..An attempt should be made to determine what is in Fraction A.
- No information on whether there were any biological replicates of the samples? It is difficult to make reliable conclusions based on one.
- The authors claim that D-trp appears as a result of fermentation. However, it was not verified whether Hidakakombu contained D-trp before fermentation.
Discussion needs to be revised.
Line 237 - Potentially, because its effect on hypertension has not been experimentally proven.
Section 4.2 This is the appropriate place to refer to the probiotic effect of the tested product mentioned in the title. There is no discussion of this issue and it lacks references to other literature data related to this topic.
Section 4.3 - The first sentence is redundant.
Line 277-278 Khedr et al. (Eur J Nutr (2018) 57:907–915) have previously reported the ACE-inhibiting potential and effects on vascular function of tryptophan-containing peptides.
Section 4.3 - the whole discussion is based on the identified tryptophan. This description is fine, but what is missing is that it nevertheless represents a very small percentage of the total aqueous extract of L. casei 001-fermented Hidakakombu and in < 3 kDa fraction, which had the highest ACE inhibitory effect, 99% constituted other compounds (which, in addition to tryptophan, certainly also contribute to this effect).
Conclusions
Line 302-303 "Lactic acid bacteria are widely accepted as a familiar food"? - LAB are rather not a food by themselves, just a supplement to foodproducts.
Line 306-308 Sentence should be reframed. In general, I suggest reconsidering the conclusions.
References are not prepared in accordance with the guidelines; Foods requires a format: Journal Articles:
1. Author 1, A.B.; Author 2, C.D. Title of the article. Abbreviated Journal Name Year, Volume, page range.
Reviewer 2 Report
The article Antihypertensive and probiotic effects of Hidakakombu (Saccharina angustata) fermented by Lactobacillus casei 001by Sekine et al has some minor errors that must be corrected.
Antihypertensive and probiotic effects of Hidakakombu (Saccharina
angustata) fermented by Lactobacillus casei 001
Tetsuya Sekine , Hiroshi Nagai and Naoko Hamada-Sato
P1 line 16 The higher the grade, the higher the price, but the lower the grade, the lower the price
of the kelp. = rephrase into more scientific/professional language
P1 line 32 to adapt to into adapted to
P1 line 34 and 35 the protein composition and amino acid sequence of seaweeds are different
from those of terrestrial organisms into with different protein composition and amino acid
sequence from those of terrestrial organisms
P2 line 47 and 48 Anti-tumor activity [5] and anti-viral activity [6] have been reported
as health functionalities of Hidakakombu. Combine this sentence...it look very strange to stand
alone after previous one.
P2 line 96 The main idea of this paper is to add value to lower grade Hidakakombu...what grade
did you use in the experiments? What is the origin of L. casei 001? Did you isolated and
determined it and maintained in your laboratory or it was bought from culture banks?
P3 line 128 hydatid...define what is hydatid
P3 line 133 BCP plate counting agar... plate counting agar is different growth media and cannot be
used together with BCP like you stated. Rephrase.
P4 line 180 graphs are wrong...A should be B and vice versa
P11 line 318 recheck references and use abbreviation according to instructions to authors
P11 line 396 International Journal of Tryptophan Research…use abbreviation
P11 line 412 Journal of Food Measurement and Characterization…use abbreviation

Round 2
Reviewer 1 Report
Ad 9 - Information in introduction still does not provide an adequate background.
The sentences added to the introduction ("This study was focused on the fermentation of kelp with lactic acid bacteria. The antihypertensive effect enhanced by fermentation and the probiotic effect added by fermentation were examined for adding value to low-grade Hidakakombu") do not give a better background to the study, because they do not refer to literature data but only repeat information about what was addressed in the study. When mentioning probiotics, there is no indication of whether kelp is commonly fermented by LAB and consumed as a product with probiotic effect.
Ad 12 -For the future, I encourage you to use cryoprotectants to maintain high viability of microorganisms after lyophilisation.
Ad 23- I still believe that for ACE inhibitory activity assays at least 3 replicates of the test sample is the standard required for the results to be reliable. The fact that kelp has been fermented hundreds of times has no relation to whether tests need to have replicates.
Ad 24 and 26 - The discussion section has changed very little and is still not well described.
